# Usefulness of T2-Weighted Images with Deep-Learning-Based Reconstruction in Nasal Cartilage

**DOI:** 10.3390/diagnostics13193044

**Published:** 2023-09-25

**Authors:** Yufan Gao, Weiyin (Vivian) Liu, Liang Li, Changsheng Liu, Yunfei Zha

**Affiliations:** 1Department of Radiology, Renmin Hospital of Wuhan University, Wuhan 430060, China; 2MR Research, GE Healthcare, Beijing 100176, China

**Keywords:** nasal cartilage, rhinoplasty, deep learning, magnetic resonance imaging

## Abstract

Objective: This study aims to evaluate the feasibility of visualizing nasal cartilage using deep-learning-based reconstruction (DLR) fast spin-echo (FSE) imaging in comparison to three-dimensional fast spoiled gradient-echo (3D FSPGR) images. Materials and Methods: This retrospective study included 190 set images of 38 participants, including axial T1- and T2-weighted FSE images using DLR (T1WI_DL_ and T2WI_DL_, belong to FSE_DL_) and without using DLR (T1WI_O_ and T2WI_O_, belong to FSE_O_) and 3D FSPGR images. Subjective evaluation (overall image quality, noise, contrast, artifacts, and identification of anatomical structures) was independently conducted by two radiologists. Objective evaluation including signal-to-noise ratio (SNR) and contrast-to-noise ratio (CNR) was conducted using manual region-of-interest (ROI)-based analysis. Coefficient of variation (CV) and Bland–Altman plots were used to demonstrate the intra-rater repeatability of measurements for cartilage thickness on five different images. Results: Both qualitative and quantitative results confirmed superior FSE_DL_ to 3D FSPGR images (both *p* < 0.05), improving the diagnosis confidence of the observers. Lower lateral cartilage (LLC), upper lateral cartilage (ULC), and septal cartilage (SP) were relatively well delineated on the T2WI_DL_, while 3D FSPGR showed poorly on the septal cartilage. For the repeatability of cartilage thickness measurements, T2WI_DL_ showed the highest intra-observer (%CV = 8.7% for SP, 9.5% for ULC, and 9.7% for LLC) agreements. In addition, the acquisition time for T1WI_DL_ and T2WI_DL_ was respectively reduced by 14.2% to 29% compared to 3D FSPGR (both *p* < 0.05). Conclusions: Two-dimensional equivalent-thin-slice T1- and T2-weighted images using DLR showed better image quality and shorter scan time than 3D FSPGR and conventional construction images in nasal cartilages. The anatomical details were preserved without losing clinical performance on diagnosis and prognosis, especially for pre-rhinoplasty planning.

## 1. Introduction

The nose encompasses seven primary anatomical components, including the paired nasal bones, upper lateral cartilages (ULC), lower lateral cartilages (LLC), and the septum [1,2]. Surgical reconstruction or nasal implantation is the first choice for defects and deformities of nasal cartilage caused by trauma and diseases [3,4]. The nasal septum cartilage (SP) is the main component of the nasal septum. Aesthetic and functional improvement for severe septal deformities and “crippled” septal plates often necessitates a septoplasty procedure for proper reshaping [5]. Additionally, nasal deformities secondary to cleft lip and palate, such as disproportionate nostril size, can be treated with secondary correction by excision, replication, or augmentation of the LLC [6]. Careful removal of partial ULC can effectively reduce the width of the lateral nasal tip in patients who have wide middle and lower thirds of the nose [7]. Preoperative planning demands delineated nasal cartilage morphology and anatomy. Contrast-enhanced high-resolution micro-CT provides more than 10-time resolution of MRI cartilage images (36.8 μm for micro-CT and 450 μm for MRI), but it is limited to ex vivo studies [8]. Applying CT images to identify the anatomical relations of the dorsal septum and the anterior cranial prevents individuals from possible complications [9]. However, CT lacks sufficient contrast and information of soft tissues and also has ionizing radiation, leading to controversial issues about its utility [8,10,11]. To combine CT and MR images could reduce the average operation time by 20 min via establishing a patient-tailored three-dimensional (3D) nose model before rhinoplasty to assist pre-surgical planning [12].

MRI is the most effective tool for the display of soft tissues including cartilages, particularly nasal cartilage [13,14,15]. However, there is only one clinical MRI study in application of the spoiled-gradient-echo-based sequence without fat saturation with spatial resolution of 0.9 mm × 0.9 mm × 1 mm and acquisition time of approximately 5 min in nasal cartilage [15], and the protocol has been subsequently used in clinics [12,16]. There have been few MRI studies of the improvement of image quality for nasal cartilage despite its importance in diagnostic value and pre-surgery planning [12]. In contrast, numerous convolutional neural network (CNN) and neural network (NN) architectures have been utilized to improve image quality, target labeling, and tissue segmentation. For example, an application of deep convolution neural networks (DCNNs) removes banding artifacts on brain and knee MR images to obtain higher SNR and consistent signal changes [17]. DCNNs in combination with residual learning and multi-channel strategy on brain MR images showed well-robust denoising performance [18]. DCNNs with up to 11-fold under-sampling could remove artifacts and maintain fine anatomy on two-dimensional cardiac magnetic resonance (MR) images. In other words, deep learning networks can efficiently and accurately segment, position, and label lesions such as colon [19], brain [20], liver [21], and anatomical structures (e.g., proximal femur [22] and vestibule [23]). Thus, it plays an important role in assisting clinical diagnosis and treatment selection.

Thin-slice spin-echo (SE) magnetic resonance imaging has the potential in pre-surgery planning for patients with unilateral cleft lip–nose–palate (UCLP) and can monitor the morphological changes of implants for post-surgical and secondary correction [14,24]. Although a 3D MRI sequence has the advantages of high signal-to-noise (SNR) and less partial volume effect, long scan time may introduce motion artifacts that greatly reduce the image quality of the nasal cartilage. Two-dimensional thin-slice images usually have a lower SNR, but they might achieve a balance between SNR and acquisition time using deep-learning-based reconstruction (DLR). An inline optimized CNN algorithm in the MR scanner is a data-driven end-to-end method to remove noise and Gibb’s artifact in the process of image reconstruction and can further retain and purify structure details without extra time for reconstruction and even with shorter acquisition time [25,26].

In this study, we aimed to investigate the clinical value of DLR-based high-resolution and short-acquisition-time fast spin-echo (FSE) sequences in the display of the morphological nasal cartilage using three-dimensional fast spoiled gradient echo (3D FSPGR) as reference.

## 2. Materials and Methods

### 2.1. Patients

This retrospective single-center study was approved by our institutional review board (No. WDRY2022-K274). Our study was conducted in accordance with the Declaration of Helsinki and its amendments. From June to November 2022, patients at the age of 18 years and older attending our hospital had clinical indications for maxillofacial MRI but no metal implants or contraindications for MRI. Informed consent was waived because of the retrospective nature of the study. Patients who had nasal trauma or tumors or incomplete MRI data were also excluded. The flowchart of patients’ enrollment and exclusion is shown in Figure 1.

### 2.2. MRI Acquisition

All axial T1- and T2-FSE images and 3D FSPGR images were acquired on a single 3.0-T MRI system (SIGNA™ Architect, GE-Healthcare, Milwaukee, WI, USA) with a 19-channel combined head-neck coil. Two-dimensional T1- and T2-weighted images using both conventional (labeled as T1WIo and T1WIo, respectively, and belong to FSEo) and deep-learning-based reconstruction algorithm (labeled as T1WI_DL_ and T1WI_DL_, respectively, and belong to FSE_DL_). The commercial name of DLR is AIR™ Recon DL (GE Healthcare) [27]. The deeply optimized CNN algorithm was embedded in the MR image reconstruction pipeline to directly input original k-space data and output DLR-based images with less noise and truncation artifacts to achieve shorter scan time and higher image quality. This inline post-processing DLR performs two functions within the MR image reconstruction pipeline: ringing reduction and SNR elevation [27,28]. MRI sequences and imaging parameters are shown in Table 1.

### 2.3. Evaluation of Images

#### 2.3.1. Qualitative Image Analysis

All 190 sets of images in DICOM format from 38 participants were transferred to an advanced workstation (GE Healthcare), and patient information was anonymized. Two radiologists (Reader 1 and Reader 2 with 1 and 7 years of experience in imaging diagnosis, respectively) were blinded to image information and independently reviewed and assessed subjective quality of these five series. Qualitative image quality (including noise, contrast, artifacts, and overall image quality) was evaluated using a five-point Likert scale (1 = unacceptable, hindering diagnosis; 2 = poor; 3 = moderate; 4 = good; 5 = excellent). Identification of anatomical structures (including the septal cartilage =SP, upper lateral cartilage = ULC, and lower lateral cartilage = LLC) was also assessed using another five-point Likert scale (5, clearly identified; 4, there is image quality distortion, but does not hinder the identification of anatomical structures; 3, slightly hindering the identification of anatomical structures; 2, severe image distortion, hindering the identification of anatomical structures; 1, unable to identify anatomical structures).

#### 2.3.2. Quantitative Image Analysis

Each sequence was anonymized in order to avoid bias. To quantify SNR and contrast-to-noise ratio (CNR), three circle regions of interest (ROI) with a minimum area of 1 cm^2^ were respectively placed on the lower lateral cartilage, proximal superficial fat, and background on five sets of images by Reader 2 in avoidance of cavities and regions with obvious artifacts or abnormal signal areas, and then, the measurements were repeated 3 times and average to one to be reported. The signal intensity of each ROI was used to compute SNR and CNR with the following formulas:SNR = SI_cartilage_/SD_background_
CNR = |SI_cartilage_ − SI _superficial fat_|/SD_background_
where SI represents mean signal intensity, while SD represents signal intensity standard deviation.

### 2.4. Thickness of Nasal Cartilage

The thickness of the lower lateral, upper lateral, and septal part of cartilage was manually measured at the anterior and inferior portion of SP, the medial crus of LLC, and midpoint of ULC [2,29] on five series of image sets in 3D Slicer software [30]. Measurements were repeated three times to validate consistency of measurements.

### 2.5. Statistical Analysis

The ordered categorical variables are reported in median and interquartile ranges (IQRs), while the continuous variables are reported in mean and standard deviation (SD). Inter-modality comparisons of image quality were performed using the paired *t*-test or Wilcoxon signed-rank test between DLR and original MR images. To assess the repeatability of cartilage thickness measurements across various protocols, we calculated the mean, SD, and percent coefficient of variation (%CV). Additionally, Bland–Altman plots were utilized to determine the mean difference and 95% limits of agreement (LOA) for cartilage thickness measurements (in millimeters). The inter-rater agreement of image quality indexes was evaluated by kappa statistics (κ < 0.21: poor, κ = 0.21–0.40: fair, κ = 0.41–0.60: moderate, κ = 0.61–0.80: good, and κ = 0.81–1.00: excellent). *p* < 0.05 was considered as a statistically significant difference. All statistical analyses were performed using SPSS Statistics version 26.0 (IBM).

## 3. Results

### 3.1. Patient Characteristics

A total of 38 patients (10 men and 28 women, mean age = 25.5 ± 2.3) with complete MRI datasets were included after excluding 9 for nasal trauma or tumors and 4 for incomplete MRI data among 51 volunteers. Characteristics of the study patients and indications for maxillofacial MRI are shown in Table 2. Figure 2 illustrates the MR images of a patient who underwent augmentation rhinoplasty with autogenous ear cartilage graft after one year.

### 3.2. Scan Time

The mean scan times for T1WI_DL_, T2WI_DL_, and 3D FSPGR are 302 ± 28 s, 245 ± 18 s, and 345 ± 23 s, respectively. The mean scan times for T1WI_DL_ and T2WI_DL_ were respectively reduced by 14.2% and 29% compared to 3D FSPGR (*p* = 0.0376 and 0.0014 < 0.05, respectively).

### 3.3. Qualitative Analysis

The results showed a significant difference of qualitative characteristics (noise, contrast, and artifacts) between FSE_DL_ (including T1WI_DL_ and T2WI_DL_) and FSE_O_ (including T1WI_O_ and T2WI_O_), as well as between FSE_DL_ and 3D FSPGR (Both *p* < 0.001). Figure 3 illustrates different slices of image sets for one pre-rhinoplasty 25-year-old female patient. FSE_DL_ images showed less noise and artifacts, as well as better contrast (*p* < 0.001), compared to FSEo and 3D FSPGR. The overall image quality between T1WI_DL_ and T2WI_DL_ showed no significant difference (*p* = 0.655). Identification of anatomical structures on FSE_DL_ was significantly better than on FSE_O_ and 3D FSPGR (all *p* < 0.001). Anatomy display of LLC and SP was optimal on T2WI_DL_ followed by T1WI_DL_ and 3D FSPGR, while no significantly different display of ULC were observed for T1WI_DL_ and 3D FSPGR (*p* > 0.05). Inter-rater agreement of qualitative assessment was fair to excellent (all κ > 0.61, Table 3). The mean scores of image quality for all series of two raters are presented in Figure 4.

### 3.4. Quantitative Image Analysis

Axial FSE quantitative analysis showed significantly higher SNR and CNR on FSE_DL_ (including T1WI_DL_ and T2WI_DL_) images compared to FSE_O_ (including T1WI_O_ and T2WI_O_) and 3D FSPGR images (all *p* < 0.001). As shown in Table 4, T2WI_DL_ showed the highest mean SNR and CNR among any other image sets (all *p* < 0.05).

### 3.5. Thickness of Nasal Cartilage

The mean thicknesses (mm) of SP, LLC, and ULC measured on 3D FSPGR (1.65 ± 0.29, 0.77 ± 0.13, and 0.81 ± 0.11, respectively) were smaller than T2WI_DL_ (1.81 ± 0.10, 0.95 ± 0.07 mm, and 0.93 ± 0.08 mm, respectively), and the difference was statistically significant. In addition, 3D FSPGR showed a greater percent coefficient of variation (%CV). Three corresponding %CV values of cartilages in order showed more consistent thickness measurements on T2WI_DL_ (5.3%, 7.1%, and 8.4%) and T1WI_DL_ (8.7%, 9.5%, and 9.7%) than 3D FSPGR images (17.7%, 17.1%, and 13.5%) in Table 5. The Bland–Altman plots present the consistency of cartilaginous thickness between any two image sets (T2WI_DL_, T1WI_DL_, and 3D FSPGR) in Figure 5.

## 4. Discussion

Two-dimensional equivalent-thin-slice nasal cartilage T1- and T2-weighted MR images using DL algorithm revealed better pre-surgery morphological structures and higher image quality than 3D FSPGR and conventional reconstruction images. Additionally, 2D imaging had shorter scan time compared to 3D acquisition in this study. Diagnosis and prognosis performed equivalently and even better for no loss of structure details—especially providing more anatomical information of pre-rhinoplasty planning on T2WI_DL_.

Surgical reconstruction or implantation of the nasal cartilage are required when trauma or diseases cause defects and deformities of nasal cartilage; moreover, autologous septal cartilage is also accepted as the standard nasal grafting material [31]. An advanced imaging method is mandatory to assess the preoperative anatomy of a patient when counseling a doctor for operating planning [10,32]. Three-dimensional facial bone computed tomography has been used to analyze the development of the nasal septum and measurement of the harvestable septal cartilage [33]. However, biases such as the overlapping between septal cartilage and alar cartilage, the overlapping of keystone areas, and anatomical variations all make the use of CT for rhinoplasty operations limited. MRI could be an alternative tool for this purpose, but there is no standard or routine MRI protocol for nasal cartilage. Additionally, long scan time brings involuntary and respiratory motion artifacts, degrades image quality and largely hinders the clinical applications of nasal cartilage MRI. Conventionally, both T1WI and T2WI were often used in clinics to reveal the edge of cartilage, abnormalities in cartilages, and surrounding tissues [24,32]. Intricacies and constraints of alar cartilage in living individuals demand shorter MR scan time, for long times bring blurred images caused by involuntary movements of the nostrils and fail in assessing cartilage thickness, surface area, and volume compared to scans on a cadaver [15]. In addition, the presence of stripe motion artifacts on the FSE images is mostly ascribed to unavoidable movements such as respiration [28]. Therefore, imaging acceleration is an essential key factor to minimize motion-induced image deterioration. The total acquisition time of the DLR images in our study was reduced by 26.6% compared to the 3D FSPGR, and FSE_DL_ even showed improved image quality, similar to other FSE_DL_ applications in anatomical districts [25,26] (e.g., spine) with up to 70% reduction in total acquisition time and no different frequency of major findings, overall image quality, or diagnostic confidence.

Rhinoplasty, such as preservation rhinoplasty (PR), demands the precise stripping of nasal cartilage from soft tissues. Protection of nasal cartilage needs attention for the maintenance of the keystone area and dorsal aesthetic lines [34]. Pre-surgical imaging is helpful in delineating anatomical information (e.g., position of nasal cartilage and surrounding soft tissues). In our study, DLR 2D sequences showed improved image quality such as sharpness and contrast. Surprisingly, T2WI_DL_ instead of T1WI_DL_ showed the best image quality, especially contrast for ULC and surrounding tissues on pre-surgical FSE_DL_ and anatomical details, compared to 3D FSPGR (*p* < 0.001). In addition to the utility of autogenous septal cartilage grafts and the septoplasty procedure, common techniques used in previous studies for nasal tip modification involved reshaping the lower lateral cartilage [6]. In fact, an increasing number of rhinoplasty surgeons have reported the usefulness of the upper lateral cartilage modification [7,35]. However, aggressive resection of the ULC without accurate preoperative evaluation carries risks of internal nasal valve disruption, along with prodigious resection of the LLC, which greatly affects the function of the external nasal valve [35]. T2WI_DL_ imaging provides an optimal pre-surgical imaging method for surgeons to estimate the cut area when ULC is preserved as much as possible to maintain the anatomical and functional relationship between the ULC and the septum.

Consistent with a previous study reported by Visscher et al. [15], T1WI_O_ showed the worst image quality compared to 3D FSPGR (*p* < 0.001). Conventionally, 3D sequences possess better spatial resolution and lower partial volume effect than 2D sequences and are more sensitive to inhomogeneities of the main magnetic field, and faster signal decay during readout might cause obvious blurring when a long echo train length is used [36]. In our study, the volume difference (0.01 mm^3^) between an isotropic resolution of 3D FSPGR (0.04^3^ mm^3^) and conventional reconstruction 2D FSE (0.03^3^ mm^3^) resulted in a drop of SNR and CNR to 18.7% and 72%, respectively, presumably deteriorating the detectability of anatomical details and lesions. SNR, resolution, and scan time (triangle balance) dominate the resultant MR image contrast [37,38]. However, DLR 2D FSE-based sequences broke the concept of triangle balance and outperformed original images in terms of both quantitative and qualitative evaluation. In accordance to a deep-learning-based denoising application of thin-slice high-resolution 2D fat-suppressed proton density-weighted image (FS-PDWI) for the knee joint presenting more useful multiple planar reformation (MPR) than 3D FS-PDWI [39], T1WI_DL_ and T2WI_DL_ showed better image quality than 3D SFPGR T1WI in our study. This newly commercial inline deep learning reconstruction algorithm that we used in this study has been also utilized to improve image quality and diagnostic performance on the heart [40,41], prostate [42], central nervous system, [43] and peripheral nerve [44] via elevating signal-to-noise ratio (SNR) and contrast to-noise ratio (CNR), as well as removing Gibb’s artifacts [41,45]. It has great advantages of removing so-called non-useful information (e.g., noise) before image reconstruction via a 10,000-kernal CNN model that employs no bias terms and rectified linear unit (ReLU) activations to identify 4.4 million features on directly received image data immediately after scanning on a computer equipped with a tensor processing unit (TPU).

In our study, coefficient of variation and Bland–Altman plots were used to compare the repeatability of five different images. The intra-observer agreements for cartilage thickness measurements were significantly improved and benefited from the better contrast of images compared to FSE_O_ images. Poorly sketching blurry cartilage edges on FSE_O_ and 3D FSPGR led to underestimated thicknesses of cartilages and relatively low intra-observer agreements of objective assessments on nasal cartilages. In contrast, FSE_DL_ was interchangeable with standard FSE_O_ and 3D FSPGR for equal or less than 10% CV of thickness measurements and the almost perfect intra-observer agreements, contributing at least an equivalent diagnosis efficacy (e.g., no loss of small pathologic findings, particularly in low-SNR regions or artifacts [46]). For clinical practices, it would reduce intra-rater biases for follow-ups.

Our study had several limitations. First, the small sample size (*n* = 38) and monocentric design might limit our findings to generalization. Second, this is a single MRI scanner and single hospital study. The diagnostic efficacy of this technology should be also evaluated on 1.5-T and 3.0-T MRIs. Finally, further applications in routine clinical practices, like post-surgical monitoring, should be explored. All the above would be included in the subsequent studies for encouraging preliminary clinical results of the DL-based reconstruction technique in nasal cartilage MRI in this present study.

In conclusion, deep-learning-based reconstruction FSE nasal MR imaging, especially for upper lateral cartilage, improved image quality, reduced scan time up to 29%, possessed better diagnostic performance, and maximized image information retention.

## Figures and Tables

**Figure 1 diagnostics-13-03044-f001:**
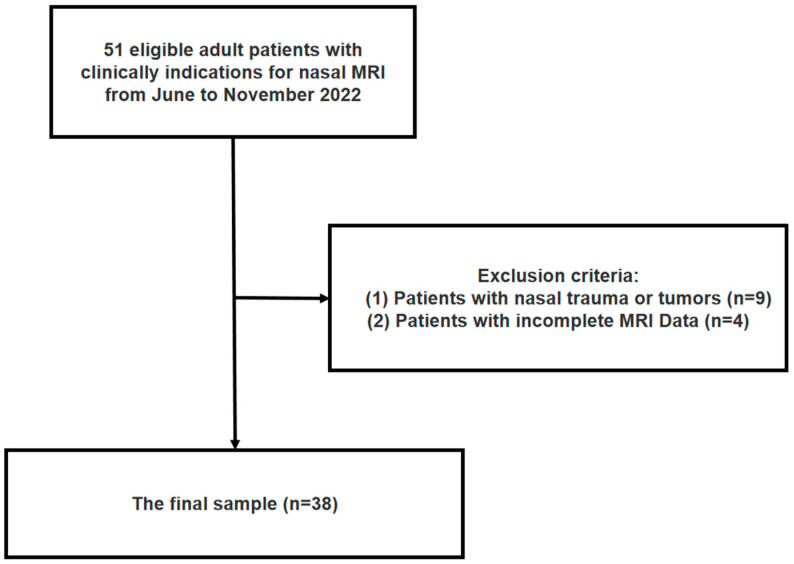
Flowchart of patient enrollment and exclusions.

**Figure 2 diagnostics-13-03044-f002:**
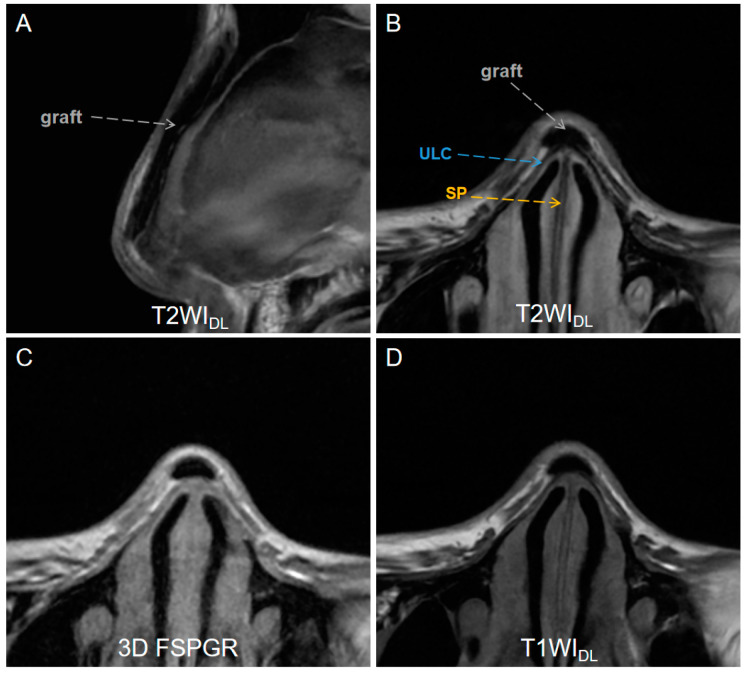
A low-signal-intensity graft (gray dashed arrows) on (**A**) sagittal T2WI_DL_ image (DLR T2-weighted FSE images), (**B**) axial T2WI_DL_ image, (**C**) axial 3D FSPGR image (three-dimensional fast spoiled gradient-recalled images), (**D**) T2WI_O_ (original T2-weighted FSE images), and (**D**) axial T1WI_DL_ (deep learning–reconstructed T1-weighted FSE images) image of a 26-year-old female who underwent an MR examination one year after receiving augmentation rhinoplasty with autogenous ear cartilage. T2WI_DL_ shows a clearer boundary between the graft and the tissue. DLR = deep-learning-based reconstruction, SP = septal cartilage (yellow dashed arrow), ULC = upper lateral cartilage (blue dashed arrow).

**Figure 3 diagnostics-13-03044-f003:**
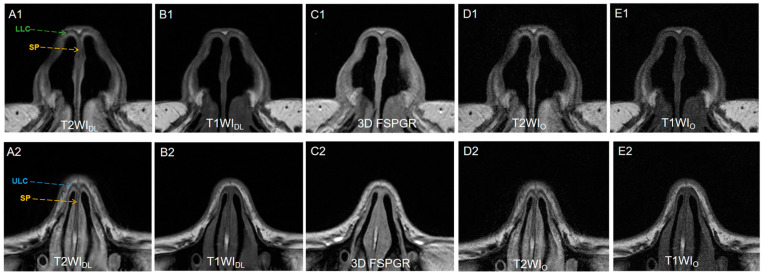
(**A1**,**A2**) T2WI_DL_ (DLR T2-weighted FSE images), (**B1**,**B2**) T1WI_DL_ (DLR T1-weighted FSE images), (**C1**,**C2**) 3D FSPGR (three-dimensional fast spoiled gradient-recalled images), (**D1**,**D2**) T2WI_O_ (original T2-weighted FSE images), and (**E1**,**E2**) T1WI_O_ (original T1-weighted FSE images) in axial view of a 25-year-old female nasal cartilage. The overall image quality and contrast of FSE_DL_ images showed better than any other image sets because of less noise. T2WI_DL_ showed the best anatomical structure of nasal cartilage for higher contrast. No significantly different display of ULC between T1WI_DL_ and 3D FSPGR was observed (*p *> 0.05) while 3D FSPGR showed relatively poor image quality of SP. DLR = deep-learning-based reconstruction, SP = septal cartilage (yellow dashed arrow), LLC = lower lateral cartilage (green dashed arrow) and ULC = upper lateral cartilage (blue dashed arrow).

**Figure 4 diagnostics-13-03044-f004:**
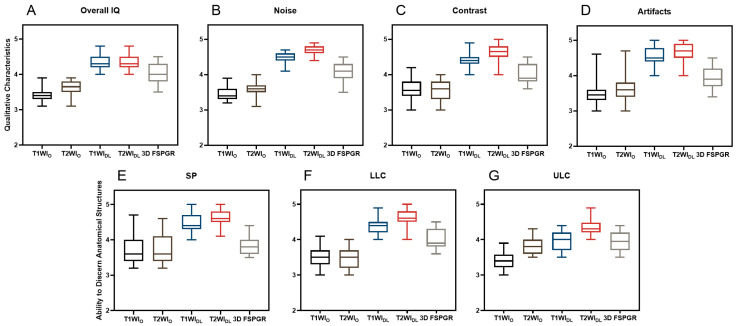
Qualitative assessment including (**A**) overall IQ, (**B**) noise, (**C**) contrast, (**D**) artifacts, and ability to discriminate anatomical structures such as (**E**) SP, (**F**) LLC, (**G**) UCL of T1WI_O_, T1WI_DL_, T2WI_O_, T2WI_DL_, and 3D FSPGR images for a total of 38 patients by two radiologists were displayed in median and the first and third one-fourth quartile in the box plots. T1WI_O_ = original T1-weighted FSE images, T1WI_DL_ = deep learning–reconstructed T1-weighted FSE images, T2WI_O_ = original T1-weighted FSE images, T2WI_DL_ = deep learning–reconstructed T2-weighted FSE images, 3D FSPGR = three-dimensional fast spoiled gradient-recalled images. Overall IQ = overall image quality, SP = septal cartilage, ULC = upper lateral cartilage and LLC = lower lateral cartilage.

**Figure 5 diagnostics-13-03044-f005:**
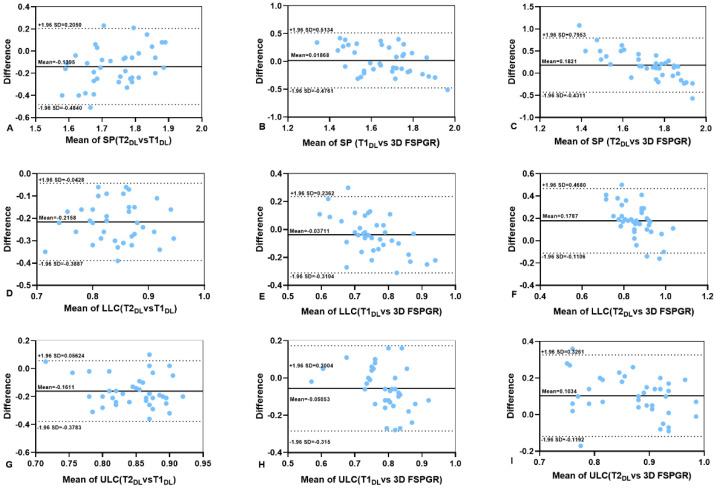
(**A**–**I**) Bland–Altman analyses of the differences between the thicknesses of cartilage measured on T1WI_DL_, T2WI_DL_, and 3D FSPGR. The upper and lower dashed black lines are limits of agreement and the full black lines are mean differences between images. T1WI_O_ = original T1-weighted FSE images, T1WI_DL_ = deep learning–reconstructed T1-weighted FSE images, T2WI_O_ = original T1-weighted FSE images, T2WI_DL_ = deep learning–reconstructed T2-weighted FSE images, 3D FSPGR = three-dimensional fast spoiled gradient-recalled images. SP = septal cartilage, ULP = upper lateral cartilage and LLC = lower lateral cartilage.

**Table 1 diagnostics-13-03044-t001:** MRI sequences and imaging parameters.

	T1WI_O_/T1WI_DL_	T2WI_O_/T2WI_DL_	3D FSPGR
TR (ms)	497	3203	7.8
TE (ms)	14.3	62.1	2.5
Thickness (mm)	2	2	2
FOV (mm × mm)	14 × 14	14 × 14	14 × 14
Recon DL Strength	Off/High	Off/High	/
Matrix	480 × 480	416 × 416	320 × 320
Acquisition time (min:s)	5:02	4:05	5:45

Note. Recon DL Strength allows users to select a deep-learning-based reconstruction strength on demand; “Off” means that only the original image is generated; “High” represents the highest SNR improvement for deep-learning-based reconstruction. TR = repetition time, TE = echo time, FOV = field of view. T1WI_O_ = original T1-weighted FSE images, T1WI_DL_ = deep learning–reconstructed T1-weighted FSE images, T2WI_O_ = original T1-weighted FSE images, T2WI_DL_ = deep learning–reconstructed T2-weighted FSE images, 3D FSPGR = three-dimensional fast spoiled gradient-recalled images.

**Table 2 diagnostics-13-03044-t002:** Characteristics of Patients and Indications for Maxillofacial MRI.

Characteristic	
Age (y)	25 ± 2
Median age	26
Gender (Male/Female)	10/28
Indication for Maxillofacial MRI	Number of subjects
Pre-rhinoplasty	24
Post-rhinoplasty	3
UCLP	2
Deviated septum	9

Note. Except age shown in mean ± deviation standard, and other characteristics data were shown in numbers of participants. UCLP = unilateral cleft lip–nose–palate.

**Table 3 diagnostics-13-03044-t003:** Inter-rater agreement of different image sets.

Reader 1 vs. Reader 2	T_1_WI_O_	T_1_WI_DL_	T_2_WI_O_	T_2_WI_DL_	3D FSPGR
	κ	κ	κ	κ	κ
overall image quality	0.7	0.8	0.8	0.8	0.8
noise	0.8	0.7	0.7	0.8	0.7
contrast	0.9	0.8	0.8	0.7	0.7
artifact	0.8	0.8	0.9	0.8	0.6
septal cartilage	0.8	0.6	0.9	0.7	0.7
upper lateral cartilage	0.7	0.7	0.9	0.8	0.7
lower lateral cartilage	0.7	0.6	0.6	0.7	0.6

Note. The inter-rater agreement of image quality indices was evaluated by kappa statistics, κ = *Kappa* values, 0.00–0.20, 0.21–0.40, 0.41–0.60, 0.61–0.80, and 0.81–1.00 indicated poor, fair, moderate, good, and excellent agreement, respectively. T1WI_O_ = original T1-weighted FSE images, T1WI_DL_ = deep learning–reconstructed T1-weighted FSE images, T2WI_O_ = original T1-weighted FSE images, T2WI_DL_ = deep learning–reconstructed T2-weighted FSE images, 3D FSPGR = three-dimensional fast spoiled gradient-recalled images.

**Table 4 diagnostics-13-03044-t004:** Qualitative image analysis.

	SNR	CNR
3D FSPGR	64.99 ± 24.25	5.80 ± 4.56
T1WI_DL_	55.23 ± 20.06	9.91 ± 9.45
vs. T1WI_O_ (*p* value)	<0.001	<0.001
vs. 3D FSPGR (*p* value)	0.082	0.213
T2WI_DL_	79.01 ± 25.72	20.66 ± 12.94
vs. T1WI_O_ (*p* value)	<0.001	<0.001
vs. 3D FSPGR (*p* value)	0.005	<0.001
T1WI_O_	14.18 ± 5.82	2.69 ± 2.05
vs. 3D FSPGR (*p* value)	<0.001	0.011
T2WI_O_	18.54 ± 2.81	3.86 ± 2.32
vs. 3D FSPGR (*p* value)	<0.001	0.101

Note. The results of quantitative analysis (including SNR and CNR) measured on T1WI, T2WI, and 3D FSPGR images were compared between deep-learning-based reconstruction images and conventional reconstruction images, deep-learning-based reconstruction images and 3D FSPGR images, and conventional reconstruction images and 3D FSPGR images.T1WI_O_ = original T1-weighted FSE images, T1WI_DL_ = deep learning–reconstructed T1-weighted FSE images, T2WI_O_ = original T1-weighted FSE images, T2WI_DL_ = deep learning–reconstructed T2-weighted FSE images, 3D FSPGR = three-dimensional fast spoiled gradient-recalled images. SNR = signal-to-noise ratio, CNR = contrast to-noise ratio.

**Table 5 diagnostics-13-03044-t005:** Thickness of nasal cartilage measured on different images.

	SP		LLC		ULC	
	Mean ± SD	%CV	Mean ± SD	%CV	Mean ± SD	%CV
3D FSPGR	1.63 ± 0.29	17.70%	0.77 ± 0.13	17.10%	0.81 ± 0.11	13.50%
T1WI_DL_	1.67 ± 0.15	8.70%	0.73 ± 0.07	9.50%	0.76 ± 0.07	9.70%
vs. T1WI_O_ (*p* value)	**0.007**		0.653		**<0.001**	
vs. 3D FSPGR (*p* value)	0.428		0.109		**0.02**	
T2WI_DL_	1.81 ± 0.10	5.30%	0.95 ± 0.07	7.10%	0.93 ± 0.08	8.40%
vs. T1WI_O_ (*p* value)	**<0.001**		0.917		0.979	
vs. 3D FSPGR (*p* value)	**<0.001**		**<0.001**		**<0.001**	
T1WI_O_	1.56 ± 0.18	11.50%	0.72 ± 0.11	15.40%	0.67 ± 0.09	13.80%
vs. 3D FSPGR (*p* value)	0.201		0.094		**<0.001**	
T2WI_O_	1.70 ± 0.16	9.10%	0.95 ± 0.13	13.50%	0.93 ± 0.10	10.80%
vs. 3D FSPGR (*p* value)	0.169		**<0.001**		**<0.001**	

Note. The result of the thickness measured on the axial T1WI, T2WI, and 3D FSPGR images were compared between deep-learning-based reconstruction images and conventional reconstruction images, deep-learning-based reconstruction images and 3D FSPGR images, and conventional reconstruction images and 3D FSPGR images. T1WI_O_ = original T1-weighted FSE images, T1WI_DL_= deep learning–reconstructed T1-weighted FSE images, T2WI_O_ = original T1-weighted FSE images, T2WI_DL_ = deep learning–reconstructed T2-weighted FSE images, 3D FSPGR= three-dimensional fast spoiled gradient-recalled images. SD = standard deviation; %CV = percent coefficient of variation (SD/mean). SP = septal cartilage, ULC = upper lateral cartilage and LLC = lower lateral cartilage. Significant *p* values are expressed in bold.

## Data Availability

The data presented in this study are available on request from the corresponding author.

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
