# Peer review of "Usefulness of T2-Weighted Images with Deep-Learning-Based Reconstruction in Nasal Cartilage"

_diagnostics, 2023, doi:10.3390/diagnostics13193044_

Round 1
Reviewer 1 Report
Comments and Suggestions for Authors
This paper aims to investigate the clinical value of DLR-based high-resolution and short-acquisition-time FSE sequences in displaying morphological nasal cartilage using three-dimensional fast spoiled gradient echo (FSPGR) based T1WI as a reference.
Overall, this is a succinct and well-focused paper. However, there are areas where the redaction could be improved to facilitate reading. It is essential to ensure that every acronym is defined upon its first mention. Additionally, providing clear terminology for the five series used is essential. Pay careful attention to the titles and legends for Tables and Figures to enhance clarity.
The Introduction is adequate, but it would be beneficial to include some information about the nasal structures (ULC/LLC). This additional context would help readers understand why these structures were chosen for the Methods section.
The Methods section is described in sufficient detail.
However, the presentation of Results could be improved. Consider presenting scan time and qualitative analysis using a graph. The graph in Figure 4 should be reformatted, as it is currently inadequate to place all three series in the same vertical line while mixing triangles, squares, and circles.
The Discussion is the weakest part of the paper. Despite the acknowledgment of the limitations, there is a lack of in-depth analysis regarding the implications of these results for clinical practice or technical development.
Comments on the Quality of English LanguageTypos and redaction.
Author Response
Response to Reviewer 1 Comments
Thank you very much for taking the time to review this manuscript. Please find the detailed responses below and the corresponding corrections in the re-submitted files.
Point-by-point response to Comments and Suggestions for Authors:
- Overall, this is a succinct and well-focused paper. However, there are areas where the redaction could be improved to facilitate reading. It is essential to ensure that every acronym is defined upon its first mention.
Response: With regard to acronyms, we have integrated all inconsistent acronyms in the first draft, carefully checked every acronym throughout the article, and also ensure that each abbreviation is used in its full name when it is first mentioned.
- Additionally, providing clear terminology for the five series used is essential. Response: Thank you for your valuable suggestion. We have revised abstract and method sections in the manuscript and checked each terminology of the five series in the context (Line 15-18 of Page 1, Line 118-122 of Page 3). In this study, axial T1- and T2-weighted FSE images using DLR was labeled as T1WIDL and T2WIDL, and original reconstructed images without using DLR was labeled as T1WIO and T2WIO. Besides, they belong to FSEDL/FSEO.
- Pay careful attention to the titles and legends for Tables and Figures to enhance clarity.
Response: To enhance the readability and clarity of our manuscript, we have added the suggestions in supplementary materials and modified the titles and legends of relevant Tables and Figures. The corresponding corrections were also modified in the resubmitted manuscript.
- The Introduction is adequate, but it would be beneficial to include some information about the nasal structures (ULC/LLC). This additional context would help readers understand why these structures were chosen for the Methods section.
Response: The nose encompasses seven primary anatomical components, the paired nasal bones, upper lateral cartilages (ULC), lower lateral cartilages (LLC), and the septum [1, 2]. Surgical reconstruction or nasal implantation is the first choice for defects and deformities of nasal cartilage caused by trauma and disease-caused invasions[3, 4]. The nasal septum cartilage (SP) is the main component of the nasal septum in supporting of the nasal cavity and midface. Aesthetic and functional improvement for severe septal deformities and "crippled" septal plates, often necessitates a septoplasty procedure for proper reshaping[5]. Additionally, nasal deformities secondary to cleft lip and palate, such as disproportionate nostril size, can be treated with secondary correction by excision, replication, or augmentation of the LLC[6]. Careful removal of partial ULC can effectively reduce the width of the lateral nasal tip in patients who have wide middle and lower thirds of the nose[7]. We’ve also added the abovementioned context in the Introduction section (Line 47-57 of Page 2) and relevant articles in the reference list.
- The Methods section is described in sufficient detail. However, the presentation of Results could be improved. Consider presenting scan time and qualitative analysis using a graph. The graph in Figure 4 should be reformatted, as it is currently inadequate to place all three series in the same vertical line while mixing triangles, squares, and circles.
Response: We formatted Figure 4 in display of qualitative analysis and Table 4 in presentation of quantitative analysis.

- The Discussion is the weakest part of the paper. Despite the acknowledgment of the limitations, there is a lack of in-depth analysis regarding the implications of these results for clinical practice or technical development.
Response: We have modified Discussion section, particularly focusing on clinical practices and imaging parameters, including a comprehensive comparison with traditional preoperative imaging (Line 356-374 of Page 13-14), the advantage of effective motion artifacts elimination (Line 367-374 of Page 14),and the display of surgery-related altered anatomic structures in comparison with pre-rhinoplasty (Line 380-396 of Page 14), and corresponding articles were also added in the reference list.
reference
[1] Vila P M, Jeanpierre L M, Rizzi C J, et al. Comparison of Autologous vs Homologous Costal Cartilage Grafts in Dorsal Augmentation Rhinoplasty: A Systematic Review and Meta-analysis[J]. JAMA Otolaryngol Head Neck Surg,2020,146(4):347-354.
[2] Kim T K, Jeong J Y. Surgical anatomy for Asian rhinoplasty: Part II[J]. Arch Craniofac Surg,2020,21(3):143-155.
[3] Dong W, Xu Y, Fan F. Secondary Rhinoplasty for Unilateral Cleft Nasal Deformity[J]. Plast Reconstr Surg,2022,150(6):1352e-1354e.
[4] Cao Y, Sang S, An Y, et al. Progress of 3D Printing Techniques for Nasal Cartilage Regeneration[J]. Aesthetic Plast Surg,2022,46(2):947-964.
[5] Shah J P, Youn G M, Wei E X, et al. Septoplasty Revision Rates in Pediatric vs Adult Populations[J]. JAMA Otolaryngol Head Neck Surg,2022,148(11):1044-1050.
[6] Brown W E, Lavernia L, Bielajew B J, et al. Human nasal cartilage: Functional properties and structure-function relationships for the development of tissue engineering design criteria[J]. Acta Biomater,2023,168:113-124.
[7] Pensler J M. The role of the upper lateral cartilages in aesthetic rhinoplasty[J]. Aesthet Surg J,2009,29(4):290-294.
Reviewer 2 Report
Comments and Suggestions for Authors
The work presented on the medical application is quite interesting,
The work reflects the complete analysis of the images but it can be more interesting if the method of reconstructed images is elaborated.
Author Response
Response to Reviewer 2 Comments
Thank you very much for taking the time to review this manuscript. Please find the detailed responses below and the corresponding corrections in the re-submitted files.
Point-by-point response to Comments and Suggestions for Authors:
- The work presented on the medical application is quite interesting. The work reflects the complete analysis of the images but it can be more interesting if the method of reconstructed images is elaborated.
Response: In addition to the information about deep learning reconstruction (DLR) algorithm mentioned in the Introduction (Line 73-84 of Page 2) and Methods (Line 117-128 of Page 3) section of this manuscript, we have also added related description in details in the Discussion section. This newly-commercial inline deep learning reconstruction algorithm has been also utilized to improve image quality and diagnostic performance on heart[1, 2], prostate[3], central nervous system[4] and peripheral nerve[5] via elevating signal-to-noise ratio (SNR) and contrast to-noise ratio (CNR) as well as removing Gibb’s artifacts[2, 6]. It has great advantages of removing so-called non-useful information (e.g., noise) before image reconstruction via a 10-thousoand-kernal CNN model that employs no bias terms and Rectified Linear Unit (ReLU) activations to identify 4.4 million features on directly-received image data immediately after scanning on a computer equipped with a tensor processing unit (TPU). This description has been added at Line 408-413 of Page 13-14 as well as in the reference list.
Reference
[1] Muscogiuri G, Martini C, Gatti M, et al. Feasibility of late gadolinium enhancement (LGE) in ischemic cardiomyopathy using 2D-multisegment LGE combined with artificial intelligence reconstruction deep learning noise reduction algorithm[J]. Int J Cardiol,2021,343:164-170.
[2] van der Velde N, Hassing H C, Bakker B J, et al. Improvement of late gadolinium enhancement image quality using a deep learning-based reconstruction algorithm and its influence on myocardial scar quantification[J]. Eur Radiol,2021,31(6):3846-3855.
[3] Park J C, Park K J, Park M Y, et al. Fast T2-Weighted Imaging With Deep Learning-Based Reconstruction: Evaluation of Image Quality and Diagnostic Performance in Patients Undergoing Radical Prostatectomy[J]. J Magn Reson Imaging,2022,55(6):1735-1744.
[4] Sreekumari A, Shanbhag D, Yeo D, et al. A Deep Learning-Based Approach to Reduce Rescan and Recall Rates in Clinical MRI Examinations[J]. AJNR Am J Neuroradiol,2019,40(2):217-223.
[5] Zochowski K C, Tan E T, Argentieri E C, et al. Improvement of peripheral nerve visualization using a deep learning-based MR reconstruction algorithm[J]. Magn Reson Imaging,2022,85:186-192.
[6] Pesapane F, Codari M, Sardanelli F. Artificial intelligence in medical imaging: threat or opportunity? Radiologists again at the forefront of innovation in medicine[J]. Eur Radiol Exp,2018,2(1):35.
Reviewer 3 Report
Comments and Suggestions for Authors
Thank you for the opportunity to review this manuscript, which compares a commercial deep learning reconstruction technique, AIR, for T1 and T2 weighted MRI images to three-dimensional fast spoiled gradient-recalled (3D FSPGR) images of nasal cartilage using both qualitative and quantitative metrics. The study incorporates data from 38 patients, each imaged with all three modalities.
While the aim of the manuscript is sound and well-justified, it suffers from a multitude of errors encompassing grammar, presentation, numerical inconsistencies, and overall clarity. In its current state, the manuscript does not meet the quality standards necessary for publication. I strongly recommend a major revision, urging the authors to meticulously review each sentence and number in the manuscript to ensure accuracy before submitting for publication.
Below find my comments:
1) Manuscript Language: The manuscript is rife with grammatical errors, beginning even with the title, which should read "Reconstruction of T2-Weighted Images." I recommend thorough language editing for clarity.
2) Abstract: The objective is ambiguous. The phrase "feasibility of anatomical structures of nasal cartilage" is unclear. The abstract should explicitly state the research objective.
3) Abbreviations in Abstract: The terms "SNR" and "CNR" are not spelled out upon first usage. This needs to be clarified.
4)Abstract: Consistency in Terminology: The abstract refers to "DLR FSE MRI" in its objective and "FSEDL" in the results section. The authors should settle on a single term for the deep learning-based algorithm to avoid reader confusion.
6)Abstract: “and had greater CV”. Abbreviation CV not written out at first use. It is unclear what it means.
7)Abstract: For this section of the abstract:
“3D FSPGR, compared to T2WIDL, underestimated nasal 24 cartilage thickness (1.63mm vs. 1.81mm for SP, 0.77mm vs. 0.95mm for ULC, and 0.81mm vs. 25 0.93mm for LLC) and had greater CV (8.7%, 9.5% and 9.7% vs. 17.7%, 17.1% and 13.5%). In addition, 26 the total acquisition time was reduced by 14.2% to 29% in the FSEDL group compared to 3D FSPGR.”
it is not indicated whether these comparisons were statistically significant.
8)Abstract: “conventional construction images (FSEO) in nasal cartilages”.
In the conclusion, it is the first time that the authors mention “conventional construction images (FSEO)”. There is no FSEO mentioned in the objective or methods section of the abstract.
9)Introduction: P2L55: “higher SNR”. SNR is not written out at first use. In contrast, it is written out later in line 66 (“high signal-to-noise (SNR)”)).
10)Introduction: P2L52: “CNN/NN” is not written out at first use. In contrast, it is written out later in line 70 (“An optimized convolutional neural network (CNNs) algorithm”).
This is also the case for other abbreviations in the manuscript that I will not comment on further.
Please revise the manuscript in such a fashion that abbreviations are written out at first use and then only the abbreviation is used when it was first defined.
11)Results: P4L152.154: “An illustration example of FSEDL, FSEO and 3D FSPGR images for a patient who underwent augmentation rhinoplasty with autogenous ear cartilage graft after one year. „.
This sentence makes no sense as no figure reference is provided. Where is the example?
12)Methods/results: Abbreviations in tables and figures need to be explained in the respective figure/table legends.
13) Methods: 2.3.1.: Did readers 1 and 2, respectively, evaluate the 5 series for all 38 patients? Therefore, each reader evaluated 5x38= 190 image series (DICOM data), right? This should be mentioned so that the statistical comparison becomes clearer later on in the results section.
14) Methods: 2.3.2. Who draw the ROIs? When the two Readers draw the ROIs and you mention that it started on 3DFSPGR and was then copied to the other modalities, how can the readers be blinded to the image modality? Or were they blinded before for qualitative assessment and later on they were not blinded anymore for the quantitative assessment? This should be made clear in this section to avoid confusion.
15)results: section 3.2. contains no information about whether there was statistical significance in the differences in scan time (i.e., p-values).
16)Results: Figure 2 legend mentions A-E. But the figure includes A-F. Also, the legend wrongly states E is T1WIo although it is F which is T1WIo. Please double-check and revise all table and figure legends to avoid any mistakes.
17) Results: the title of table 5: “Table 5. of nasal cartilage.” Does not make sense. And it contains mistakes (e.g., the column title contains the term “Table 1.”). Please provide a comprehensive table title and legend including the abbreviations used in the table.
Also, for table 5, provide an extra column with the p-value indicating whether the means of SP, LLC, and ULC were statistically different between the imaging groups shown.
18) Results: section 3.4.: it mentions “In Table 3, T2WIDL showed the highest
mean SNR and CNR among any other image sets (all P < 0.05).” However, then, table 4 is shown below and not table 3.
19) Results: Table 4: the legend mentions: “†††P < 0.001 indicated 3D FSPGR in comparison to DLR images.” and “♯♯♯P < 0.001 indicated 3D FSPGR in comparison to T2WIDL images”. But T2WIDL is also DLR (deep learning reconstruction)? ††† should be “comparison to T1WIDL” not DLR.
20)Discussion/Conclusion: the manuscripts results led to the conclusion that DLR-based reconstruction with the algorithm AIR led to superior results compared to the conventional technique and 3D FSPGR. However, there is no focus in the manuscript on the algorithm AIR used here, besides some short information in the Methods section (P3L95-102).
Are there other algorithms than AIR? Did other studies use AIR for DLR and if yes what were their findings? Are there limitations to AIR? What could further improvements to DLR reconstruction algorithms be?
Comments on the Quality of English LanguageNeeds revision in grammar/abbreviations, and clarity of presentation.
Author Response
Response to Reviewer 3 Comments
Thank you very much for taking the time to review this manuscript. Please find the detailed responses below and the corresponding corrections in the re-submitted files.
Point-by-point response to Comments and Suggestions for Authors:
1)Manuscript Language: The manuscript is rife with grammatical errors, beginning even with the title, which should read "Reconstruction of T2-Weighted Images." I recommend thorough language editing for clarity.
Response: A native English speaker has helped us refine the language and improve the readability and sufficient clarity of our article. We’ve changed the original title to “Usefulness of T2-Weighted Images with Deep Learning-based Reconstruction in Nasal Cartilage”.
2) Abstract: The objective is ambiguous. The phrase "feasibility of anatomical structures of nasal cartilage" is unclear. The abstract should explicitly state the research objective.
Response: We’ve modified “Objective” to “To evaluate the feasibility of visualizing nasal cartilage using deep learning-based reconstruction (DLR) fast spin echo (FSE) imaging in comparison to three-dimensional fast spoiled gradient echo (3D FSPGR) images”. (Line 10-12 of Page 1).
3) Abbreviations in Abstract: The terms "SNR" and "CNR" are not spelled out upon first usage. This needs to be clarified.
Response: We carelessly missed the complete spelling for those terms. We have carefully checked acronyms throughout the article to ensure that each abbreviation is used in its full name when first mentioned.
4)Abstract: Consistency in Terminology: The abstract refers to "DLR FSE MRI" in its objective and "FSEDL" in the results section. The authors should settle on a single term for the deep learning-based algorithm to avoid reader confusion.
Response: We have revised Abstract and Method sections in the manuscript and checked each terminology of the five series in the context (Line 15-18 of Page 1, Line 118-122 of Page 3). In this study, axial T1- and T2-weighted FSE images using DLR was labeled as T1WIDL and T2WIDL, and original reconstructed images without using DLR was labeled as T1WIO and T2WIO. Besides, they belong to FSEDL/FSEO.
6)Abstract: “and had greater CV”. Abbreviation CV not written out at first use. It is unclear what it means.
Response: We’ve revised the manuscript with complete spelling for abbreviations at first use and only abbreviations were used afterwards. In our study, coefficient of variation (CV) was used to compare the reliability and reproducibility of five different sequences. In general, less than 10% CV shows a high degree of intra-observer reliability and inter-observer reproducibility for observer[1, 2].
7)Abstract: For this section of the abstract:
“3D FSPGR, compared to T2WIDL, underestimated nasal 24 cartilage thickness (1.63mm vs. 1.81mm for SP, 0.77mm vs. 0.95mm for ULC, and 0.81mm vs. 25 0.93mm for LLC) and had greater CV (8.7%, 9.5% and 9.7% vs. 17.7%, 17.1% and 13.5%). In addition, 26 the total acquisition time was reduced by 14.2% to 29% in the FSEDL group compared to 3D FSPGR.”
it is not indicated whether these comparisons were statistically significant.
Response: To address your concerns, we have revised both the Abstract and Method sections. Statistical analysis has been conducted for the anatomical structures using different imaging techniques (Line 37-40 of Page 1) and shown in a more clear and comprehensive way.
8)Abstract: “conventional construction images (FSEO) in nasal cartilages”.
In the conclusion, it is the first time that the authors mention “conventional construction images (FSEO)”. There is no FSEO mentioned in the objective or methods section of the abstract.
Response: In this study, axial T1- and T2-weighted FSE images using DLR was labeled as T1WIDL and T2WIDL, and original reconstructed images without using DLR was labeled as T1WIO and T2WIO. Besides, they belong to FSEDL/FSEO. The definition and meaning of the five sequences are consistent and clear after our revision (Line 16-18 of Page 1, Line 118-121 of Page 3).
9)Introduction: P2L55: “higher SNR”. SNR is not written out at first use. In contrast, it is written out later in line 66 (“high signal-to-noise (SNR)”)).
10)Introduction: P2L52: “CNN/NN” is not written out at first use. In contrast, it is written out later in line 70 (“An optimized convolutional neural network (CNNs) algorithm”).
This is also the case for other abbreviations in the manuscript that I will not comment on further.
Please revise the manuscript in such a fashion that abbreviations are written out at first use and then only the abbreviation is used when it was first defined.
Response: For Question 9) and 10), we have carefully checked acronyms throughout the article accordingly with full spelling and abbreviations for the first time and only abbreviations were used afterwards. These changes have been made in the body of our manuscript correspondingly.
11)Results: P4L152.154: “An illustration example of FSEDL, FSEO and 3D FSPGR images for a patient who underwent augmentation rhinoplasty with autogenous ear cartilage graft after one year. „.
This sentence makes no sense as no figure reference is provided. Where is the example?
Response: Figure 2 illustrates MR images of a patient who underwent augmentation rhinoplasty with autogenous ear cartilage graft after one year (Line 198-200 of Page 5). The corresponding corrections are added in the resubmitted manuscript ( Figure 2: 219-227 of Page 6 ).
12)Methods/results: Abbreviations in tables and figures need to be explained in the respective figure/table legends.
Response: We have carefully revised all the Tables and Figures to ensure that the used abbreviations were clearly elaborated in the manuscript. Additionally, legends and notes accompanying with these Tables and Figures were carefully revised as “TR = Time of Repetition, TE = time of Echo, FOV= Field of View. T1WIO = original T1-weighted FSE images, T1WIDL = deep learning–reconstructed T1-weighted FSE images, T2WIO = original T1-weighted FSE images, T2WIDL= deep learning–reconstructed T2-weighted FSE images, 3D FSPGR = three-dimensional fast spoiled gradient-recalled images.” (Table 1, Line 132-141 of Page 4), “Except age (mean ± deviation standard), data are numbers of participants. UCLP = unilateral cleft lip-nose palate.” (Table 2, Line 214-217 of Page 6), “SP= septal cartilage (yellow dashed arrow), LLC= lower lateral cartilage (green dashed arrow) and ULC= upper lateral cartilage (blue dashed arrow).” (Figure 2 and 3, Line 219-227 of Page 6 and 248-258 of Page 7), “A=overall image quality; B=noise; C=contrast; D=artifact; E=septal cartilage; F=upper lateral cartilage; G=lower lateral cartilage.” (Table 3, Line 266-272 of Page 8) and “DLR = deep learning-based reconstruction, T1WIO = original T1-weighted FSE images, T1WIDL = deep learning–reconstructed T1-weighted FSE images, T2WIO = original T1-weighted FSE images, T2WIDL= deep learning–reconstructed T2-weighted FSE images, 3D FSPGR = three-dimensional fast spoiled gradient-recalled images. Overall IQ = Overall image quality, SP = septal cartilage, ULC = upper lateral cartilage and LLC = lower lateral cartilage.” (Figure 4, Line 284-292 of Page 10; Table 4, Line 306-312 of Page 11; Figure 5, Line 324-331 of Page 12, and Table 5, Line 333-340 of Page 12-13).
13) Methods: 2.3.1.: Did readers 1 and 2, respectively, evaluate the 5 series for all 38 patients? Therefore, each reader evaluated 5x38= 190 image series (DICOM data), right? This should be mentioned so that the statistical comparison becomes clearer later on in the results section.
Response: Both Reader 1 and Reader 2 independently evaluated the five series for all 38 patients. The information on each reader assessed a total of 190 image series was added information in the Method section of the manuscript (Line 146-147 of Page 4).
14) Methods: 2.3.2. Who draws the ROIs? When the two Readers draw the ROIs and you mention that it started on 3DFSPGR and was then copied to the other modalities, how can the readers be blinded to the image modality? Or were they blinded before for qualitative assessment and later on they were not blinded anymore for the quantitative assessment? This should be made clear in this section to avoid confusion.
Response: We apologized for any confusion regarding the Method of sketching ROI for quantitative image analysis in the Method section. Only Reader 2 was responsible for sketching the ROIs, and this process was applied to all 190 images. The results were obtained by averaging the measured values from 3 separate measurements. We revised the methodology of ROI-based measurements in the Method section to better understanding (Line 158-163 of Page 4).
15)results: section 3.2. contains no information about whether there was statistical significance in the differences in scan time (i.e., p-values).
Response: To address your concerns, we have revised the Result section. Statistical analysis for significant difference of each structure were shown in Line 229-233 of Page 6-7.
16)Results: Figure 2 legend mentions A-E. But the figure includes A-F. Also, the legend wrongly states E is T1WIo although it is F which is T1WIo. Please double-check and revise all table and figure legends to avoid any mistakes.
Response: Thanks for your suggestions, all table and figure legends were carefully and thoroughly reviewed.
17) Results: the title of table 5: “Table 5. of nasal cartilage.” Does not make sense. And it contains mistakes (e.g., the column title contains the term “Table 1.”). Please provide a comprehensive table title and legend including the abbreviations used in the table.
Also, for table 5, provide an extra column with the p-value indicating whether the means of SP, LLC, and ULC were statistically different between the imaging groups shown.
Response: To address your concerns, we have corrected relevant sentences (Line 235-247 of Page 11-12) and Table 5 (Line 332-340 of Page 12-13) in the Result section.

18) Results: section 3.4.: it mentions “In Table 3, T2WIDL showed the highest
mean SNR and CNR among any other image sets (all P < 0.05).” However, then, table 4 is shown below and not table 3.
19) Results: Table 4: the legend mentions: “†††P < 0.001 indicated 3D FSPGR in comparison to DLR images.” and “♯♯♯P < 0.001 indicated 3D FSPGR in comparison to T2WIDL images”. But T2WIDL is also DLR (deep learning reconstruction)? ††† should be “comparison to T1WIDL” not DLR.
Response: For Question 18) and 19), we formatted Table 4 in display of quantitative analysis with P value as below and modified correspondingly in the body of our manuscript.

20)Discussion/Conclusion: the manuscripts results led to the conclusion that DLR-based reconstruction with the algorithm AIR led to superior results compared to the conventional technique and 3D FSPGR. However, there is no focus in the manuscript on the algorithm AIR used here, besides some short information in the Methods section (P3L95-102).
Are there other algorithms than AIR? Did other studies use AIR for DLR and if yes what were their findings? Are there limitations to AIR? What could further improvements to DLR reconstruction algorithms be?
Response: In addition to the information about deep learning reconstruction (DLR) algorithm mentioned in the Introduction (Line 73-84 of Page 2) and Methods (Line 117-128 of Page 3) section of this manuscript, we have also added related description in details in the Discussion section. This newly-commercial inline deep learning reconstruction algorithm also has been utilized to improve image quality and diagnostic performance on heart[3, 4], prostate[5], central nervous system[6] and peripheral nerve[7] via elevating signal-to-noise ratio (SNR) and contrast to-noise ratio (CNR) as well as removing Gibb’s artifacts[4, 8]. It has great advantages of removing so-called non-useful information (e.g., noise) before image reconstruction via a 10-thousoand-kernal CNN model that employs no bias terms and Rectified Linear Unit (ReLU) activations to identify 4.4 million features on directly-received image data immediately after scanning on a computer equipped with a tensor processing unit (TPU). This description has been added at Line 412-421 of Page 14-15 as well as in the reference list.
Reference
[1] Visscher D O, van Eijnatten M, Liberton N, et al. MRI and Additive Manufacturing of Nasal Alar Constructs for Patient-specific Reconstruction[J]. Sci Rep,2017,7(1):10021.
[2] Ricatti G, Veronese N, Gangai I, et al. Hoffa's fat pad thickness: a measurement method with sagittal MRI sequences[J]. Radiol Med,2021,126(6):886-893.
[3] Muscogiuri G, Martini C, Gatti M, et al. Feasibility of late gadolinium enhancement (LGE) in ischemic cardiomyopathy using 2D-multisegment LGE combined with artificial intelligence reconstruction deep learning noise reduction algorithm[J]. Int J Cardiol,2021,343:164-170.
[4] van der Velde N, Hassing H C, Bakker B J, et al. Improvement of late gadolinium enhancement image quality using a deep learning-based reconstruction algorithm and its influence on myocardial scar quantification[J]. Eur Radiol,2021,31(6):3846-3855.
[5] Park J C, Park K J, Park M Y, et al. Fast T2-Weighted Imaging With Deep Learning-Based Reconstruction: Evaluation of Image Quality and Diagnostic Performance in Patients Undergoing Radical Prostatectomy[J]. J Magn Reson Imaging,2022,55(6):1735-1744.
[6] Sreekumari A, Shanbhag D, Yeo D, et al. A Deep Learning-Based Approach to Reduce Rescan and Recall Rates in Clinical MRI Examinations[J]. AJNR Am J Neuroradiol,2019,40(2):217-223.
[7] Zochowski K C, Tan E T, Argentieri E C, et al. Improvement of peripheral nerve visualization using a deep learning-based MR reconstruction algorithm[J]. Magn Reson Imaging,2022,85:186-192.
[8] Pesapane F, Codari M, Sardanelli F. Artificial intelligence in medical imaging: threat or opportunity? Radiologists again at the forefront of innovation in medicine[J]. Eur Radiol Exp,2018,2(1):35.

Round 2
Reviewer 1 Report
Comments and Suggestions for Authors
The latest revision of this manuscript has incorporated all the requested amendments I proposed.
Author Response
Dear reviewer:
Thanks very much for your kind work and consideration on our paper. On behalf of my co-aithors, we would like to express our great appreciation to you.
Thank you and best regards.
Reviewer 3 Report
Comments and Suggestions for Authors
The authors have addressed my concerns. However, the reference list is not provided in the revised manuscript. I recommend to double-check whether the references are correct.
Comments on the Quality of English LanguageThe authors have addressed my concerns. However, the reference list is not provided in the revised manuscript. I recommend to double-check whether the references are correct.
Author Response
Thank you very much for taking the time to review this manuscript. We have carefully checked and provided the complete reference list in the re-submitted files (Line 396-527 of Page 12-15).